

# Ten genes and two topologies: an exploration of higher relationships in skipper butterflies (Hesperiidae)

Ranjit Kumar Sahoo[1], Andrew D. Warren[2], Niklas Wahlberg[3,4], Andrew V. Z. Brower[5], Vladimir A. Lukhtanov[6,7] and Ullasa Kodandaramaiah[1]

[1] School of Biology, Indian Institute of Science Education and Research Thiruvananthapuram, Thiruvananthapuram, Kerala, India
[2] McGuire Center for Lepidoptera and Biodiversity, Florida Museum of Natural History, University of Florida, UF Cultural Plaza, Gainesville, FL, USA
[3] Department of Biology, Lund University, Lund, Sweden
[4] Department of Biology, University of Turku, Turku, Finland
[5] Evolution and Ecology Group, Department of Biology, Middle Tennessee State University, Murfreesboro, TN, USA
[6] Department of Insect Systematics, Zoological Institute of Russian Academy of Sciences, St. Petersburg, Russia
[7] Department of Entomology, St. Petersburg State University, St. Petersburg, Russia

Corresponding author
Ranjit Kumar Sahoo,
sahoork@iisertvm.ac.in

## ABSTRACT

Despite multiple attempts to infer the higher-level phylogenetic relationships of skipper butterflies (Family Hesperiidae), uncertainties in the deep clade relationships persist. The most recent phylogenetic analysis included fewer than 30% of known genera and data from three gene markers. Here we reconstruct the higher-level relationships with a rich sampling of ten nuclear and mitochondrial markers (7,726 bp) from 270 genera and find two distinct but equally plausible topologies among subfamilies at the base of the tree. In one set of analyses, the nuclear markers suggest two contrasting topologies, one of which is supported by the mitochondrial dataset. However, another set of analyses suggests mito-nuclear conflict as the reason for topological incongruence. Neither topology is strongly supported, and we conclude that there is insufficient phylogenetic evidence in the molecular dataset to resolve these relationships. Nevertheless, taking morphological characters into consideration, we suggest that one of the topologies is more likely.

## INTRODUCTION

A robust phylogeny is the key to understanding historical macroevolutionary processes that have shaped extant diversity. For instance, a phylogenetic hypothesis is needed to address questions regarding patterns of morphological evolution, coevolution, and historical biogeography, as well as for a higher-level classification system. Among invertebrates, butterflies have been the most popular study systems in evolutionary biology (*Boggs, Watt & Ehrlich, 2003*). Relationships among and within butterfly families have been largely studied by phylogenetic analyses of DNA sequence data (*Campbell, Brower & Pierce, 2000*; *Caterino et al., 2001*; *Braby, Vila & Pierce, 2006*;

*Nazari, Zakharov & Sperling, 2007*; *Wahlberg et al., 2009*; *Simonsen et al., 2011*; *Heikkilä et al., 2012*; *Wahlberg et al., 2014*; *Espeland et al., 2015*). Yet, the higher-level relationships among skipper butterflies, with more than 4,000 species in about 567 genera (*Warren, Ogawa & Brower, 2008*) and representing a fifth of the world's butterfly fauna (*Hernández-Roldán et al., 2014*), are still unsatisfactorily resolved.

Until recently, the higher-level classification of the family that has been generally followed was that proposed by *Evans (1949)* based on morphological characters. However, a major problem with skipper systematics is the remarkable uniformity of morphological structure among skipper taxa, which makes phenotype-based grouping extremely challenging (*Voss, 1952*; *Warren, Ogawa & Brower, 2008*). Following multiple attempts over the last several decades (*Voss, 1952*; *Ackery, 1984*; *Scott, 1985*; *Scott & Wright, 1990*; *Chou, 1994*; *Chou, 1999*; *Mielke, 2005*), a recent study employing molecular data suggested a classification that included five subfamilies (*Warren, Ogawa & Brower, 2008*). This classification relied on analyses of one mitochondrial and two nuclear markers, a dataset of 2,085 bp. A subsequent analysis that added morphological data (49 characters) to the same molecular data led to a revised classification that included seven subfamilies (*Warren, Ogawa & Brower, 2009*).

*Warren, Ogawa & Brower (2008*, *2009)* used Maximum Parsimony analyses with nodal support estimated through Bremer Support values (*Bremer, 1994*). Despite some strongly supported monophyletic taxa being recovered, many putative higher clades were unresolved. Specifically, uncertainty remained about relationships among the major clades within the subfamily Pyrginae. Furthermore, support for relationships among the monophyletic subfamilies Heteropterinae, Trapezitinae and Hesperiinae was weak to moderate. The status of Euschemoninae as sister to rest of the family, except Coeliadinae, received very low nodal support (Bremer support = 1), although this placement is corroborated by the early developmental characters of Euschemoninae, which are similar to those of Coeliadinae and Eudaminae (*Warren, Ogawa & Brower, 2009*).

*Yuan et al. (2015)* investigated relationships among a small subset of hesperiid taxa—23 genera from China—using 1,458 bp of mitochondrial sequence data. Their Maximum Likelihood (ML) tree also indicated uncertainty in the position of Eudaminae and Pyrginae. Another study based on complete mitochondrial genomes of six skipper butterflies representing five subfamilies (sensu *Warren, Ogawa & Brower, 2009*) failed to support the monophyly of Pyrginae (*Kim et al., 2014*).

In summary, existing studies on the higher-level relationships within this speciose butterfly family have indicated significant conflicts (*Warren, Ogawa & Brower, 2008*; *Warren, Ogawa & Brower, 2009*; *Kim et al., 2014*; *Yuan et al., 2015*), and we currently lack a robust higher level phylogenetic hypothesis for evolutionary studies or a subfamily-level classification. The reasons for conflicting topologies across studies and poor nodal support could be (a) incongruence among gene trees due to incomplete lineage sorting (*Pollard et al., 2006*; *Whitfield & Lockhart, 2007*), (b) ancestral introgression (*Eckert & Carstens, 2008*), (c) differences in characteristics of the datasets used (*Nabholz et al., 2011*), (d) inadequate taxon sampling, (e) insufficient data to resolve deeper nodes

(*Wolf et al., 2002*; *Rokas et al., 2003a*) or, (f) a near-hard polytomy due to a rapid radiation (*Kodandaramaiah et al., 2010*).

In order to bring further understanding to the higher-level phylogeny of skipper butterflies, we assembled sequences of 10 gene regions from 270 genera and analyzed a 7,726 bp dataset using both parsimony and model-based tree reconstruction methods. We also compiled the complete mitochondrial genome of 15 skipper species across five subfamilies from GenBank to compare the tree from the mitochondrial genome with that of single mitochondrial and combined nuclear genes. Consistent with the existing conflict across studies (*Warren, Ogawa & Brower, 2008*; *Warren, Ogawa & Brower, 2009*; *Kim et al., 2014*; *Yuan et al., 2015*), our analyses showed conflicting topologies at the deeper nodes of the phylogeny. To understand the reasons for the uncertainty in the phylogenetic estimation, we followed an integrative approach with systematic data encoding and tree comparison.

## METHODS

### Taxon and gene sampling

Our analyses were based on 311 ingroup specimens representing 270 hesperiid genera and 12 outgroup taxa (five Papilionidae, two Hedylidae and five Pieridae). This dataset builds on the previous study by *Warren, Ogawa & Brower (2009)* that included sequences of three protein-coding genes (mtDNA COI, EF1a and wingless). We sequenced an additional part of COI and seven more genes (ArgKin, CAD, GAPDH, IDH, MDH, RpS2 and RpS5) using protocols and primers from *Wahlberg & Wheat (2008)*. A new primer pair was designed (Table S1) to amplify the gene IDH for certain taxa. We have also included 96 additional specimens representing 71 genera to the present analyses. Our taxon sampling accounts for 60–70% of the genera of Coeliadinae (six genera), Eudaminae (38 genera), Heteropterinae (seven genera), and Trapezitinae (11 genera); 40–50% of Pyrginae (73 genera) and Hesperiinae (134 genera), and 100% of Euschemoninae (one genus). The sequences for outgroups were acquired from GenBank. We included the morphological, behavioral and ecological data matrix used in *Warren, Ogawa & Brower (2009)* in certain analyses. The sequences generated during this study were deposited in GenBank. The molecular data matrix in our study comprised 7,726 characters, more than three times that of the previous dataset (*Warren, Ogawa & Brower, 2008*; *Warren, Ogawa & Brower, 2009*).

### Dataset encoding

Along with the analysis of the concatenated dataset (nt_123), we generated context specific datasets from the concatenated gene matrix for various analyses designed to identify potential sources of conflict and/or poor nodal support. In one analysis, accounting for the impact of compositional heterogeneity, we assigned ambiguity to all the sites that potentially experience synonymous change (degen_1) (*Regier et al., 2010*; *Zwick, Regier & Zwickl, 2012*). We also checked the extent of substitution saturation in each gene matrix using DAMBE v6.4.20 (*Xia, 2013*), which showed saturation in

3rd codon positions (Fig. S1). To account for substitution saturation and degeneracy, we removed the 3rd codon positions and the 1st codon positions coding for Arginine or Lysine (noLRall1 + nt2) (*Regier et al., 2008*). In further analyses, we removed the 3rd coding positions from the concatenated dataset (nt_12) or only from the mitochondrial genes (nuclear_123 + CO_12).

We also analyzed all the nuclear genes together (nuclear_123) and reconstructed multi-gene and single-gene trees for comparison. In subsequent analyses, we also combined the morphological, ecological and behavioral characters from *Warren, Ogawa & Brower (2009)* with certain molecular datasets—nt_123 and nuclear_123. In addition, we analyzed an assembly of all protein coding sequences (13 genes) from the mitochondrial genomes of 15 skipper butterflies acquired from GenBank.

## Phylogenetic analyses

We performed Maximum Parsimony analyses in TNT v.1.1 (*Goloboff, Farris & Nixon, 2008*) using 'New Technology' searches (*Goloboff, 1999*; *Nixon, 1999*) (consisting of tree fusion, sectorial search, ratchet and tree drift) with 1,000 random addition replicates; nodal supports were derived from 1,000 bootstrap replicates (*Felsenstein, 1985*). For ML and Bayesian Inference (BI) analyses, we used RAxML v8 (*Stamatakis, 2014*) and MrBayes v3.2 (*Ronquist et al., 2012*), respectively, on the XSEDE web server through the CIPRES Science gateway (*Miller, Pfeiffer & Schwartz, 2010*). For ML analyses, we used the GTR model of substitution with gamma model of rate heterogeneity (GTR+G) and different partition schemes, either gene-based or based on rates of evolution calculated by the program Tree Independent Generation of Evolutionary Rates (TIGER) (*Cummins & McInerney, 2011*). In gene-based partitions each gene was considered as a separate partition, while in TIGER-based partitions, the characters were binned together based on their rate of evolution regardless of gene origin (used as a partitioning strategy in *Rota & Wahlberg, 2012*). TIGER partitions for the dataset were derived from the program TIGER v1.02 that calculates relative rates of evolution of each site in an alignment (*Cummins & McInerney, 2011*). The data were then divided into seven partitions based on the relative rates using an algorithm developed by Tobias Malm (J. Rota, T. Malm & N. Wahlberg, 2016, unpublished data) such that the first partition consisted of the invariant and very slowly evolving sites and the last partition consisted of sites evolving very quickly. To check whether model selection had any impact on the tree reconstruction, we also performed the above ML analyses using the best fitted model from PartitionFinder v1.1.1 (*Lanfear et al., 2012*) (detail in Fig. S2). For each ML analyses, the node supports were computed from 1,000 bootstrap replicates. Single-gene matrices were analyzed with GTR+G model and the node supports were derived from 500 bootstrap replicates. For single-gene analyses, we dropped the taxa for which the corresponding gene sequence was not available. For the mitogenome analysis, we performed ML tree searches with codon-based partitions and estimated the nodal support from 100 bootstraps.

The BI analysis of the concatenated dataset with TIGER partitions was performed with a mixed model of substitution which samples all possible models in the GTR family in proportion to their posterior distributions (*Huelsenbeck, Larget & Alfaro, 2004*) as

implemented in MrBayes 3.2 (*Ronquist et al., 2012*). We assigned the gamma model of rate heterogeneity to all the partitions; the first partition was additionally assigned a proportion of invariable sites. The program MrBayes was set to estimate the base frequencies and shape parameters from the data. Two independent runs with two chains per run were performed for ~30 million generations, sampling trees every 10,000 generations. The convergence of independent runs was analyzed from the values of potential scale reduction factors (PSRF) (PSRF close to one determines convergence) (*Gelman & Rubin, 1992*); we also checked the plots of log-likelihoods and other parameters on Tracer v1.6 (*Rambaut et al., 2014*).

## Tree comparison

To investigate the differences among the trees from multiple analyses, we compared the trees (except single-gene trees) for their topological incongruence with and without the likelihood scores. While the non likelihood-based incongruence test reflects differences in branching patterns among the topologies, the likelihood-based comparison calculates the difference between competing hypotheses with distinct topologies for a given dataset (*Planet, 2006*). For the non likelihood-based tree comparison at the deep level divergences, we used a Lento plot to depict the conflict among different trees in a two-dimensional graph (*Lento et al., 1995*). We employed the approximate unbiased (AU) test (*Shimodaira, 2002*) for likelihood-based tree comparisons. To check for incongruence among gene trees, we used partitioned bremer support (PBS) analysis that examines the contribution of each gene partition to the topological support of the consensus tree (*Baker & DeSalle, 1997*).

# RESULTS

## Multilocus tree estimation

Trees from the concatenated dataset, irrespective of parsimony or ML analysis or the partition scheme used, showed identical relationships among the early branches representing the major clades (sensu *Warren, Ogawa & Brower, 2009*) (Fig. 1A)—(i) Coeliadinae was sister to the rest of the family, (ii) Pyrginae was paraphyletic, (iii) Euschemoninae was sister to Eudaminae, and (iv) Heteropterinae, Trapezitinae and Hesperiinae were all monophyletic. The topology remained unchanged even after the addition of morphological characters to the concatenated dataset.

However, the ML trees from the combined nuclear dataset (nuclear_123) showed a contrasting topology (Fig. 1B) to that of the concatenated dataset, irrespective of the partitioning scheme: Pyrginae was monophyletic and Euschemoninae was sister to rest of Hesperiidae except Coeliadinae. The topology remained unchanged with the addition of morphological characters to the dataset (nuclear_123 + morph) as well as analyses of the subsets of the combined nuclear matrix (7gene_123, 8gene_123).

These dataset-specific variations in tree topologies were consistent across different evolutionary models and also found when partition schemes from PartitionFinder were used (detail in Fig. S2).
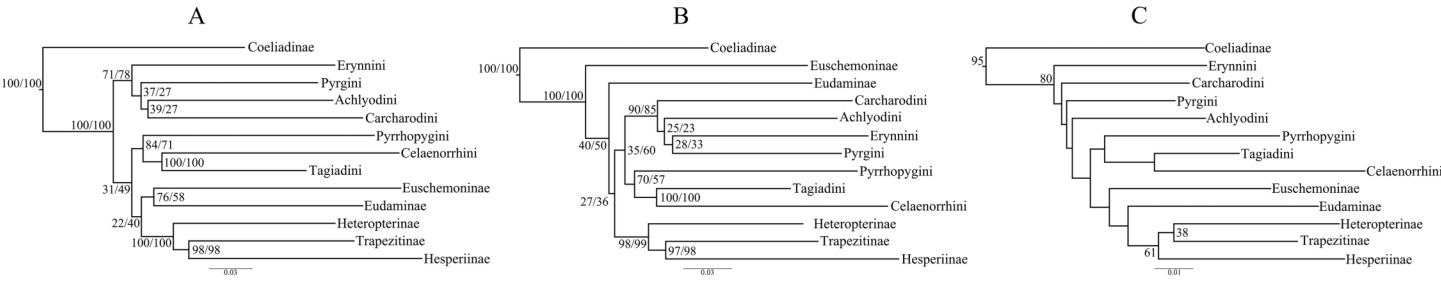

**Figure 1 Comparison of best ML trees across analyses.** The best ML trees from (A) the concatenated dataset, (B) combined nuclear dataset and (C) degenerated dataset (degen_1). The values at nodes in (A) and (B) represent support from 1,000 bootstrap trees analyzed with TIGER partitions/gene partitions. In (C), the values at the nodes are the output from the analyses using TIGER partitions only and the nodes without any value have bootstrap support <20. The other degenerated datasets (nt_12, noLRall1 + nt2) have similar or more fluctuating rearrangements for the nodes with BS < 20.

ML analyses of non-degenerated datasets (nt_12, degen_1, noLRall1 + nt2) resulted in unresolved tree topologies indicating insufficient phylogenetic signal in the 1st and 2nd codon positions of the dataset (Fig. 1C).

BI analysis of the concatenated dataset showed a topology similar to that of Fig. 1B with a few changes (detail in Fig. S3). Randomly sampled 100 trees from the MCMC generations, after discarding burnin, showed the presence of only one topology (as in Fig. 1B); however, a very low proportion (6%) of an alternate topology (as in Fig. 1A) was found in one of the runs.

## Tree comparison

To test whether multiple tree topologies across ML analyses are equally likely given the datasets, we performed an AU test (*Shimodaira, 2002*) for both the concatenated and combined nuclear datasets independently. The AU test, without any partitioning scheme, rejected (p < 0.0001) the tree topologies from the non-degenerate datasets (nt_12, degen_1, noLRall1 + nt_2). Hence, we dropped the trees from non-degenerate data sets from further analyses of tree topological similarity at higher taxonomic levels. However, two distinct topologies (Figs. 1A and 1B) were accepted as significant trees (p > 0.05) for the combined nuclear dataset, whereas only one topology (Fig. 1A) was significant (p > 0.05) for the concatenated dataset.

A visual comparison of the two distinct tree topologies (Figs. 1A and 1B) showed that all the subfamily clades (sensu *Warren, Ogawa & Brower, 2009*) except Pyrginae were monophyletic (BS > 98). Pyrginae was recovered either as monophyletic (BS = 7–60) or paraphyletic. Similarly, Euschemoninae and Eudaminae were either sisters (BS = 58–76) or non-sisters. However, due to the presence of low BS values (<76) in six out of 13 deep nodes across the analyses, we were uncertain whether there existed significant conflict among different tree topologies.

## Mito-genomic analysis

The ML tree from all 13 protein coding genes from complete mitochondrial genomes of 15 skipper butterflies had a tree-wide average BS of 80 (see Fig. S4). Along with many well-supported nodes (BS > 97), low support values (BS 30–60) were also obtained for

five out of 14 internal nodes. The lowest BS (= 32) was obtained for the node containing Eudaminae, the lineages of Pyrginae and the common ancestor of Heteropterinae and Hesperiinae. The node that placed Eudaminae as sister to one of the Pyrginae clades had BS = 42.

We found that the tree topology from the mitogenomic analysis corroborated the deep splits found in the COI gene tree; Eudaminae nested within Pyrginae rendering the latter paraphyletic, which was also the case for the concatenated dataset.

## Multiple plausible tree topologies

The presence of multiple tree topologies was not limited to the dataset-specific analyses. Regardless of the partitioning scheme or the dataset (concatenated and combined nuclear dataset) used, almost equal numbers of contrasting tree topologies were present among the ML-bootstrap trees across multiple analyses (see Fig. S5); hence, it is likely that the topology of the best ML tree is one among two equally likely topologies.

To investigate this further, we extracted 105 model-optimized trees from each of the ML analyses of the concatenated (nt_123) and combined nuclear (nuclear_123) data, separately from datasets with both partitioning schemes (gene partitions and TIGER partitions). The clades in the resulting trees were collapsed to the subfamily level except Pyrginae, which was collapsed to the level of tribes (sensu *Warren, Ogawa & Brower, 2009*), and were plotted on a Lento plot (Fig. 2) to check for support and conflict for each split. We observed conflicting splits among the trees from the combined nuclear dataset with TIGER partitions, which indicated the presence of multiple topologies; however, trees from the concatenated dataset had no conflicting splits, indicating a single topology. But the case was different when gene partitions were used—multiple topologies resulted from the concatenated dataset and only one of these topologies was recovered from the combined nuclear dataset. The AU test showed equal tree likelihoods ($p > 0.05$) for multiple topologies for both concatenated and combined nuclear datasets given the respective partition schemes as explained above.

Thus, we found two contrasting tree topologies, respectively supporting (i) monophyly of Pyrginae and non-sister status of Eudaminae and Euschemoninae, and (ii) paraphyly of Pyrginae and sister status of Eudaminae and Euschemoninae. In addition, we noticed that the position of Eudaminae with respect to Pyrginae varied across these contrasting topologies. While Eudaminae was sister to all Hesperiidae except Coeliadinae and Euschemoninae in the former, the latter topology indicated that Eudaminae was sister to the clade containing Heteropterinae, Trapezitinae and Hesperiinae.

We observed that the contrasting topologies had short branch lengths at the fluctuating clades: the interrelationship among Eudaminae, the major clades of Pyrginae and the common ancestor of Heteropterinae, Trapezitinae and Hesperiinae. The association of very low BS (<60) with these short branches suggests possible topological conflict among gene trees. To investigate this hypothesis, we examined the relationships among the conflicting clades in the individual gene trees. Because the single-gene trees had very low BS values for most of the clades, we clustered gene markers based on their topological

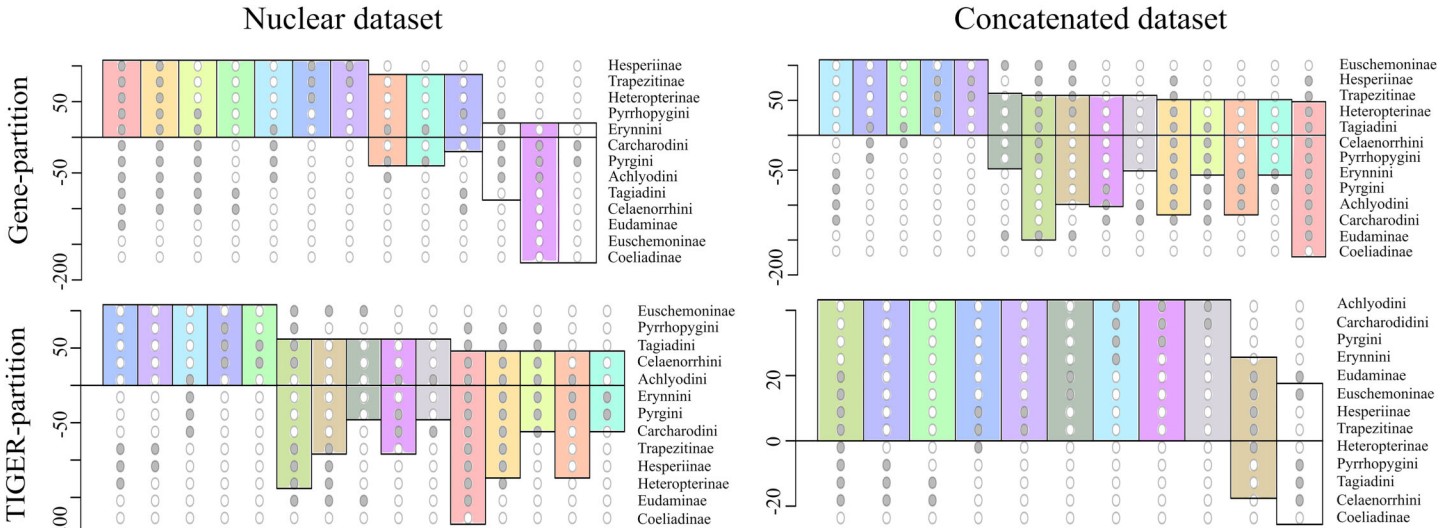

**Figure 2  Comparison of the supports/conflicts on the Lento plots.** The Lento plots were drawn from 105 ML trees recovered during best ML tree search across different datasets using two different partitioning schemes. The X-axis represents each non-trivial clade, with filled circles indicating the clade composition; the Y-axis shows relative support (values above zero) or conflict (values below zero). Splits are colour coded to aid comparison across analyses. The splits which are not coloured are found only in that particular analysis and not recovered from others.

congruency for the deeper clades and reanalyzed, expecting an improvement of nodal support and consistency in signal. For example, we combined all those gene markers from which Pyrginae was recovered as monophyletic and expected an improved BS for the Pyrginae clade in the gene cluster analysis. As expected, this gene cluster recovered Pyrginae as monophyletic with higher node support (BS = 80). However, none of the gene clusters recovered the non-sister status of Eudaminae and Euschemoninae even though their sister status was poorly supported in all the gene cluster analyses (Fig. S6). This pattern of incongruence is not an artifact of missing data in our dataset, because sequential removal of taxa with 80–40% of missing data from the analyses changed neither the topology nor the support values at deeper nodes (Table S2); however, such sequential removal gradually reduced the proportion of alternate tree topologies in the tree set from best ML tree search (Fig. S7). Hence, we could not accept the hypothesis that the two contrasting tree topologies are due to incongruence in gene histories; this is also evident from PBS analysis where no pattern of incongruence among gene trees were observed (Fig. S8).

## DISCUSSION

With a dataset of 7,726 bp from 270 hesperiid genera, we present the most comprehensive phylogeny of this important group of butterflies. Our analyses suggested that there are two contrasting topologies for the higher-level skipper phylogeny. First, we reconstructed the phylogenetic trees using the concatenated and combined nuclear datasets; the resulting trees were well-supported for higher-level relationships except at certain deep nodes. Tree comparisons revealed that there are multiple tree topologies for the relationships among major skipper lineages. We explicitly investigated gene-specific

signals for the relationships among major clades, clustered them based on their topological congruence and reanalyzed to check for consistency.

## Conflicting topologies?

Our analyses indicate the occurrence of two equally likely deep tree topologies (Figs. 1A and 1B). Interestingly, the proximate reasons for the occurrence of these contrasting topologies appear to vary depending on the partitioning scheme used for the analysis. However, neither topology was strongly supported in any analysis and the results from our explorations of incongruence among gene histories were not conclusive. Gene cluster analyses improved the nodal support for the monophyly of Pyrginae but were unable to recover the sister status of Eudaminae and Euschemoninae with good support. Similarly, from the PBS analysis, no pattern of incongruence in gene histories was observed. We conclude that there is insufficient information in the molecular dataset to resolve these relationships despite the extensive taxonomic sampling and large number of molecular characters.

The presence of conflicting topologies has also been reported from many other studies across plants (*Soltis, Soltis & Zanis, 2002*; *Burleigh & Mathews, 2004*; *Ruhfel et al., 2014*) and animals (*Rokas et al., 2003a*; *Song et al., 2012*). The possible reasons for topological incongruency are phylogenetic noise or conflict among gene trees (*Smith et al., 2015*). In case of the former, a concatenation approach is expected to give a better result (*Rokas et al., 2003b*; *Smith et al., 2015*). In the latter case, where the conflict is presumed to be a result of gene flow across taxa or incomplete lineage sorting, coalescence based methods have been used for tree reconstruction (*Jarvis et al., 2014*; *Xi et al., 2014*; *Smith et al., 2015*). However, there was very low node supports across gene trees, indicating that strong conflict across genes does not explain the patterns found here. We predict that a phylogenomic approach would provide a better outlook to this conflicting scenario or resolve the phylogeny, as such an approach has proved instrumental in other studies (*Dunn et al., 2008*; *Kocot et al., 2011*; *Smith et al., 2011*; *Johnson et al., 2013*; *Jarvis et al., 2014*; *Richart, Hayashi & Hedin, 2016*).

## Systematic implications

Our study confirmed that all the subfamilies, possibly except Pyrginae, are monophyletic and received high BS support across multiple analyses. We are uncertain about the monophyly of Pyrginae, as our study reveals homoplastic character distributions that could potentially be explained by the occurrence of ancestral introgression among its early lineages. Hence, when a certain combination of genes was used for phylogenetic construction, Pyrginae was recovered monophyletic (e.g., Fig. 1B), as in the previous study (*Warren, Ogawa & Brower, 2009*). This result appears to be supported by morphology. Similarly, we remain uncertain about the true relationships among Eudaminae, Pyrginae and the clade containing Heteropterinae, Trapezitinae and Hesperiinae, due to short branches that may be explained by their rapid divergence from each other and possibly an introgression between Eudaminae and Euschemoninae. However, the arrangement of Euschemoninae as sister to all Hesperiidae (except Coeliadinae), and Eudaminae as sister to

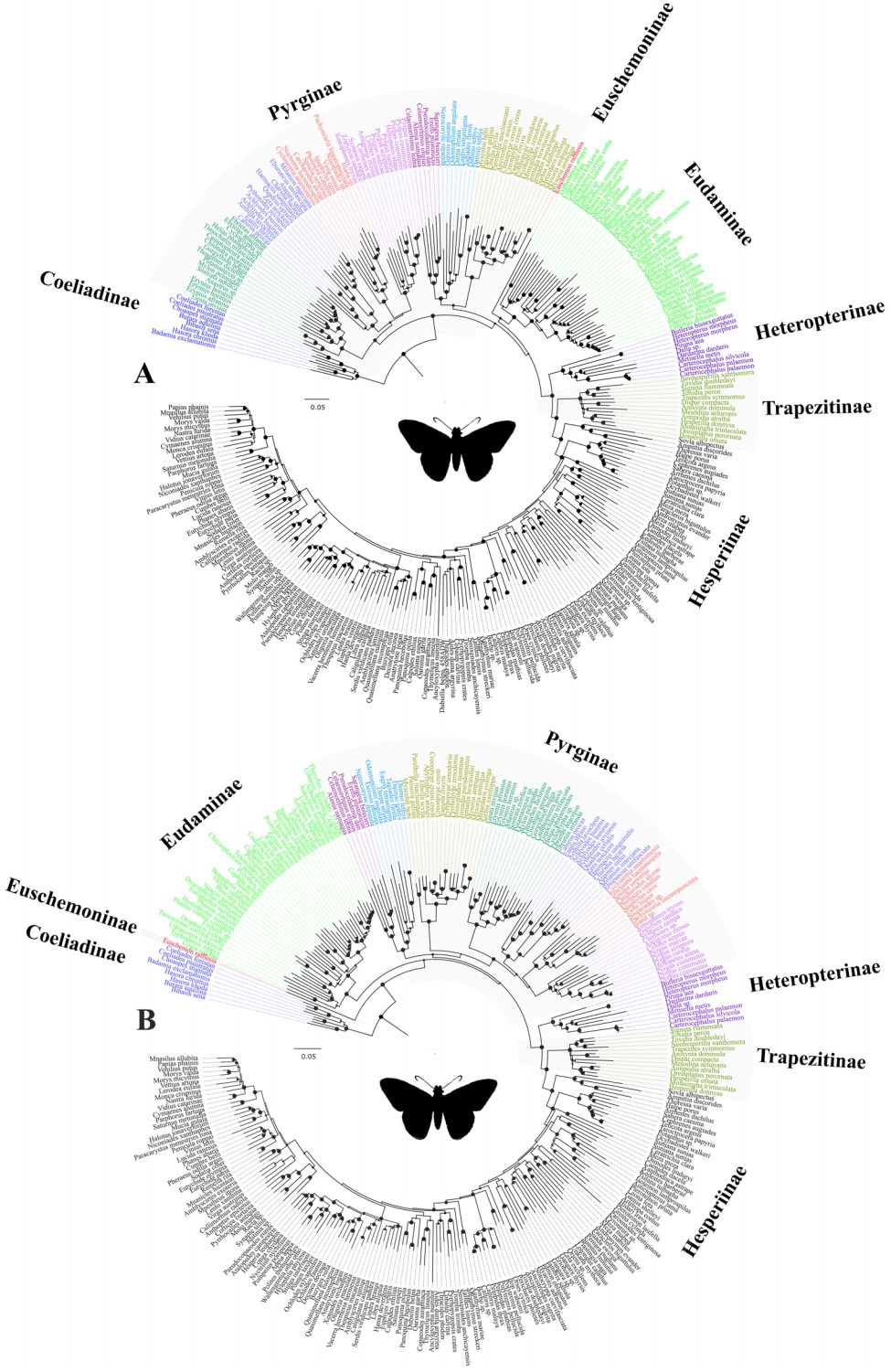

**Figure 3 The ML trees from a reduced dataset.** The ML trees from the analyses of (A) the concatenated dataset with gene partitions, and (B) the combined nuclear dataset with gene partitions. *Clito, Eracon* and three additional taxa were removed prior to the analyses (see text for detail). The size of the circle at the node corresponds to the bootstrap support that was derived from 1,000 pseudo-replicates. All taxa are colour coded based on their subfamily status, except the taxa within subfamily Pyrginae which are coloured based on their tribe. Silhouette from PhyloPic (http://phylopic.org/).

Pyrginae, is generally supported by morphology. We suggest that the relationships shown in Fig. 1B (also in Fig. 3B) should be used as the preferred phylogenetic hypothesis until a better-resolved phylogeny is available.

In addition, we observed unexpected placement of a few taxa within Pyrginae. For instance, *Eracon*, which was previously classified under Pyrgini (*Warren, Ogawa & Brower, 2009*), was found herein to group with Achlyodini. Likewise, *Clito* grouped within either Pyrgini or Erynnini based on dataset specific analyses. Moreover, we note that both *Clito* and *Eracon* sequences in our dataset have >70% missing sites. Hence, it is likely that the presence of insufficient informative sites within these taxa might influence their true positions in the phylogeny (*Wiens, 2003*; *Wiens & Morrill, 2011*). Therefore, for systematic implications, we pruned *Clito*, *Eracon* and three additional taxa with a large percentage of missing data from the dataset and reanalysed with gene partitions (Fig. 3). We observed no change in tree topology or node support values as a result of pruning these taxa. Hence, although they may appear on the tree in unorthodox positions, it is unlikely that presence of these taxa has any impact on our interpretation of higher-level relationships in the dataset as a whole.

We observed that the genus *Cabirus*, previously included within Eudaminae, grouped within Achlyodini (subfamily Pyrginae). Further study of the morphology of *Cabirus* is needed to corroborate this placement, although its position outside of Eudaminae seems to be correct. Three tribes within Hesperiinae—Aeromachini, Taractrocerini and Baorini—are monophyletic with high BS values. However, we are uncertain about the phylogenetic status of other proposed tribes within Hesperiinae due to prevalence of low BS values along the short internal branches. This indicates the possible occurrences of rapid ancestral radiation within Hesperiinae and needs further investigation.

## CONCLUSIONS

With a broad coverage of all known subfamilies, we present the higher-level relationships among skipper butterflies. Our analyses suggest possible conflicting topologies with respect to (i) monophyly or paraphyly of Pyrginae and (ii) sister or non-sister status of Eudaminae and Euschemoninae. However, none of the topologies resulting from our alternative analyses is strongly supported, and incongruences in signal among genes cannot satisfactorily resolve these differences. We surmise that there is insufficient phylogenetic information in the current dataset to resolve these relationships. It is unlikely that adding data from a few more genes will improve the results, but data from entire genomes may result in a better-resolved phylogeny. However, taking morphological characters into consideration, we suggest one of the topologies as most likely (Figs. 1B and 3B), and that this topology will aid in future studies on this group.

### Funding

The project was funded by the Department of Science and Technology (DST-RFBR-P-155) and INSPIRE Faculty Award to Ullasa Kodandaramaiah, IISER Thiruvananthapuram

and the Russian Foundation for Basic Research. Vladimir A. Lukhtanov was supported by the grant N 14-14-00541 from the Russian Science Foundation. Ranjit Kumar Sahoo was supported by a research fellowship from Council of Scientific and Industrial Research, India. The funders had no role in study design, data collection and analysis, decision to publish, or preparation of the manuscript.

### Grant Disclosures

The following grant information was disclosed by the authors:
Department of Science and Technology: DST-RFBR-P-155.
The Russian Science Foundation: 14-14-00541.
Council of Scientific and Industrial Research, India.

### Competing Interests

The authors declare that they have no competing interests.

### Author Contributions

- Ranjit Kumar Sahoo conceived and designed the experiments, performed the experiments, analyzed the data, wrote the paper, prepared figures and/or tables, reviewed drafts of the paper.
- Andrew D. Warren conceived and designed the experiments, performed the experiments, analyzed the data, contributed reagents/materials/analysis tools, wrote the paper, reviewed drafts of the paper.
- Niklas Wahlberg conceived and designed the experiments, analyzed the data, contributed reagents/materials/analysis tools, wrote the paper, reviewed drafts of the paper.
- Andrew V.Z. Brower analyzed the data, contributed reagents/materials/analysis tools, wrote the paper, reviewed drafts of the paper.
- Vladimir A. Lukhtanov conceived and designed the experiments, contributed reagents/materials/analysis tools, wrote the paper, reviewed drafts of the paper.
- Ullasa Kodandaramaiah conceived and designed the experiments, performed the experiments, analyzed the data, contributed reagents/materials/analysis tools, wrote the paper, reviewed drafts of the paper.

### DNA Deposition

The following information was supplied regarding the deposition of DNA sequences:
GenBank: KX947011–KX947012, KX947014–KX947266, KY014330–KY014419, KY019648–KY020025, KY027462–KY028742, KY045507–KY045764.

### Data Deposition

The raw data has been supplied as Supplemental Dataset Files.

### Supplemental Information

Supplemental information for this article can be found online at http://dx.doi.org/10.7717/peerj.2653#supplemental-information.

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
