# Peer review of "Ten genes and two topologies: an exploration of higher relationships in skipper butterflies (Hesperiidae)"

_PeerJ, doi:10.7717/peerj.2653_

## Round 0.1 · original submission · Major Revisions

Both reviewers agree in that the ms is a good piece of work and worth publishing. However, reviewer 2 find some major questions regarding methodologies that I think the authors must consider. Specially the way in which the partitioning schemes have been performed, the reviewer proposes some new analyses that I think will be interesting and add a value to the ms.
So, please read carefully both reviews, give answer to all the reviewers points either following their advice or explaining why you do not think the changes appropiate.

·

Basic reporting

Manuscript by Sahoo et al. presents an updated higher phylogeny of butterflies of the family Hesperiidae. This study differs from the previous attempts in a higher number of markers (10 genes) as well as in higher number of samples (271 genera). Surprisingly, such amount of information did not change much the view of the two previous studies and it did not solve the situation of the two alternative scenarios (Eudaminae basal to all except Coeliadinae or a part of Pyrginae switching their position). The authors found that the problem lies in a conflict between nuclear and mitochondrial information, however, the explanation is not, at least according the authors, caused by incongruence in gene histories. The authors showed very interesting study full of computational exercises. The text is well written, the design and the taxa are well selected. The results of the study will form a basal framework for anybody trying to study in detail any group within the family and it will enable to use a calibrated tree for phylogeographic or macro-ecological works.

Experimental design

The design of the work is correct as the authors use the up-to-date methods, they try to have as small as possible the number of missing values.

Validity of the findings

For sure the work is worth of publishing, despite the conclusion that it is impossible to reveal the true relationships due to insufficient information in the molecular dataset. The authors showed that even the dataset with large number of samples and large number of genetic markers is unable to show a single scenario as two different scenarios are equally supported. However, using different methods of the data manipulation, they showed that one of the two scenarios is more likely. With the current methodological limitations, the results are valid.

Additional comments

There are few omissions and I feel that their incorporation can improve the manuscript. First of all, the problematic of two equally good but conflicting alternatives occur also in other studies not related to these butterflies. I would be happy if the authors discuss similar situation in other organisms to make their findings more general. As the second point, I would like to see one more supplemental table showing a classic table with details of individual samples, their taxonomic hierarchy and their origin. And finally, I am unable to understand the Table S2. For sure it needs more explanation and it looks like that it has missing first column.

Reviewer 2 ·

Basic reporting

In this study, Sahoo and colleagues investigate higher level relationships among skippers. The paper is well written and organized. Overall this is a nice paper, certainly very important because of the importance of this group. I have some comments on the methods. Mainly, I feel that the authors missed some important steps in their phylogenetic inference that could potentially account for the inconsistent results. The maximum likelihood and Bayesian analyses are performed with rather strangely selected priors. The methodology used to select the partitions and models of substitution are at odds with the latest developments in this field.

The title is slightly misleading. On the contrary, it seems that the ten gene fragments did not resolve the higher level relationships among skippers.

Experimental design

Line 180-182: Why not running proper saturation tests? Many programs freely available offer such possibility (e.g. Dambe).

Line 201: gene-based partitioning is usually a very poorly performing scheme. I wonder why this was used here. The authors could (should?) have run PartitionFinder to find simultaneously the partitions and corresponding models of evolution.

Line 201-209: I am not sure I understand the TIGER analyses. Traditionally, the TIGER and kmeans algorithms can be run from PartitionFinder and then the entire partitioning scheme can be easily implemented in MrBayes or RAxML. What is the rationale to lump the different clusters recovered in “superclusters”? This seems at odds with the theoretical framework of the method (see Frandsen et al. 2015 in BMC Evol Biol).

Validity of the findings

No Comments

Additional comments

Line 95: Here as well I would suggest the use of a less radical word. Maybe “have been largely studied” instead of “largely resolved”.

Line 124-125: What is this reanalysis? Is it a maximum likelihood analysis? This sentence has a strange placement and should be moved to the result section. When looking at Figure S1, some bootstrap values are equal or close to zero. Is there a good reason for such a poor resolution?

Line 158: I would be good to have the proportion of missing data in the dataset. The figure S5 is very hard to understand. What is the “maximum proportion of missing data per taxa”? This is very strange way to present the information relative to gene sampling. I think it would be good if the authors could give a table with the taxonomic coverage for each gene fragment. This would be more helpful than Figure S5.

Line 194-197: I wonder what would be the rationale to continue using parsimony-based methods when its drawbacks are well-known when relying on molecular data.

Line 210: Which ML analysis is that?

Line 216: Why not using a more traditional approach as well (e.g. PartitionFinder)?

Line 216-217: This needs to be described more explicitly. The authors are using a reversible jump MCMC as developed by Huelsenbeck et al. (2004, Mol Biol Evol). Are the authors assuming a simple GTR model? With such a complex assemblage of gene fragments I would suspect that a gamma distributed rate variation among site and/or a proportion of invariant sites would be a better fit. This could be easily tested in PartitionFinder.

Line 220: 10 million generations is a very limited number of cycles for such a complex analysis, I am surprised that all parameters converged properly with such a small exploration of parameter space. A graph showing MCMC chain mixing or likelihood plateaus would be interesting.

Line 351-364: This is a repetition of the results section.

Line 389-390: Considering that the phylogeny of skippers remains unresolved to some degree, maybe macroevolutionary studies should be placed on the backburner for the time being and instead efforts directed toward the resolution of the skipper tree of life using next-gen data?

Line 398-400: This should in the methods section no in the discussion.

Line 416-418: Just as the title, I wonder how this sentence could possibly be supported by the results of the study. It basically says exactly the opposite. The rest of the conclusion section is also contradicting to some degree. A little streamlining is required here as well.

I think this is a very nice study, with an amazing dataset. Based on their result, the authors conclude that the phylogeny of skippers will likely need a genomic approach to be resolved, yet their study shed a new light on skipper phylogenetic relationships. My only concerns are relative to the phylogenetic analyses that I find a little strangely arranged and performed. Some reanalysis or additional analyses would be helpful and make the results more robust I believe. The TNT analysis should be dropped. The use of PartitionFinder to find a partitioning scheme and corresponding substitution models would be nice. Also the way TIGER is used in this paper is not usual, and a proper analysis using the partitioning scheme given by this algorithm along with the kmeans algorithm implemented in PartitionFinder would be nice instead of an “a posteriori” lumping of sites. Running these two analyses in ML and BI would greatly improve the study because at the moment I cannot be sure that the conflicting topologies are not the results of the rather uncommon analyses performed.

---

## Round 0.2 · Minor Revisions

The two reviewers agree in that the new ms has dealt with most of their concerns. However, reviewer 2 still finds three points that thinks need to be changed. I agree in all of them. Dambe results must be included, I see no reason to leave them behind, and also in that the new paragraph added needs some changes, since it is true that the type of gene-tree analyses done in this work is not exactly what will be needed to treat problematic cases as those concerning incomplete lineage sorting, etc. Finally, I also agree in that if some analyses have been done, even if they are only shown in the supplementary material, it is important to minimally discuss them on the main text.

So, please have into account this reviewer comments and submit again the ms, I'll myself now take the decision once I have got your new ms and answers.

·

Basic reporting

The manuscript by Sahoo et al. is an updated version of earlier submission after revision. The authors upgraded the manuscript according my comments and responded to my questions. I am satisfied by all the additions and answers and I do not have further comments nor suggestions. I looked to the responses to second referee and even there I see the answers of the authors as appropriate.

Experimental design

I explained the validity of the study design in my previous review. There were no changes in the design since the last version.

Validity of the findings

I explained the validity of the findings in my previous review. No changes in the findings was made in the updated manuscript.

Additional comments

No comments

Reviewer 2 ·

Basic reporting

I believe the authors made a lot of good changes to the manuscript. I am mostly happy with the answers to my comments. Only a few things:

- In the new paragraph suggested by another reviewer, the authors claim that they tried both concatenation and what they call “gene-tree” approaches. I don’t see how reconstructing gene trees is helping to resolve a phylogeny. If there are some gene tree discordance issues then approaches such a species-tree coalescent inference are possibly helpful, but such analyses were not conducted in this paper.

- The results of the DAMBE analyses should be included regardless of the additional “complexity” it would result in. If some analyses were conducted based on this, then there is no reason to leave the rationale out of it.

- “However, based on our prior experience we preferred gene-based partition over the partition schemes from PartitionFinder; because large number of very small-sized partitions tend to reduce the nodal support values”. This sentence is subjective and inaccurate at best. If the authors want to claim that the use of additional partitions reduces nodal support they should not only justify it in their answer but also in their manuscript based on published simulation and/or empirical studies. The results of the new PartitionFinder analysis should also be discussed in the main text.

Overall I believe the paper is improved and is almost ready for publication in PeerJ, I do not believe there is a need for rereview if the few comments above are implemented.

Experimental design

No Comments

Validity of the findings

No Comments

Additional comments

No Comments

---

## Round 0.3 · accepted · Accept

The ms is now accepted, however, I have found a mistake in one of the sentences you have added now "However, all there was very low node supports across gene trees, indicating that strong conflict across genes does not explain the patterns found here. " This sentence sounds very strange, I suppose that the "all" at the beginning of the sentence must be removed, but I prefer the authors to tell about it. So please correct this sentence while in production.